# Dynamics of the Hadley circulation in an axisymmetric model undergoing stratification periodic forcing

Nazario Tartaglione

Uni Research Climate and Bjerknes Centre for Climate Research, Bergen, Norway

*Correspondence to:* Nazario.Tartaglione@uni.no

**Abstract.** The time-dependent response of the Hadley circulation to a periodic forcing is explored via a simplified nonlinear axisymmetric model. Thermal forcing towards a given equilibrium potential temperature drives the model atmosphere. The vertical stratification of this temperature is forced to become periodically neutral with a period $t_0$. Simulations performed with values of $t_0$ ranging from 10 to 90 days exhibit stronger circulation compared to the results of a constant thermal forcing experiment. As the period increases, a transition occurs first from a stationary regime, obtained when forcing is constant, to a periodic (and possibly quasi-periodic) regime, and then to an intermittent regime, albeit one with a strong periodic component. The stream-function response to periodic forcing is generally a periodic oscillation, with two main frequencies dominating, one with a period equal or close to the forcing period and another with a period that is half of the forcing period. The former is dominant for values of $t_0$ larger than 30 days, whereas the latter is prevalent for $t_0$ smaller than 30 days. The periodic oscillations obtained in this model may be associated with the periodic oscillations observed in the tropical regions. In this case the periodic charge and discharge of moisture in the tropical atmosphere, with consequent change of stratification, may be linked to those oscillations. In the model, at forcing periods of over 63 days the response of the stream-function periodically enters into a quasi-intermittent regime, exhibiting high frequency chaotic oscillations that are modulated by the slow timescale of forcing. Sensitivity experiments for model parameters and configuration were performed to check whether results obtained are still valid under different conditions. Although for small changes of parameters the results are still valid, when parameters depart from the prescribed ones, we can observe change in the Hadley circulation dynamics.

## 1 Introduction

The Hadley cell is a specific and observable feature of tropical circulation (Dima and Wallace, 2003).The main driving process of this feature can be seen as a combination of low-latitude convective processes and interactions of tropical circulation with higher-latitude eddies (Charney, 1969). Although tropical circulation follows a seasonal cycle due to the asymmetry of solar forcing during a solar year (Fang and Tung, 1999), it is also subjected to alternating intra-seasonal strong and weak periods (Goswami and Shukla, 1984). The cycles modulate precipitation activity in the equatorial regions.

It is well known that the rainfall distribution over Indian tropical regions varies considerably from day to day with an alternating behavior associated with a few periodic or quasi-periodic cycles. Cycles of 3–7, 10–20 and 30–60 days are present in the atmospheric dynamics of tropics and subtropics. The 3–7 day cycle is related to synoptic-scale convective systems

generated over the Bay of Bengal, while the others have been associated with the cycles of monsoon rainfall over the Indian region as discussed by Kripalani et al. (2003) who showed a prevalence of the 10–20 day cycle during a normal monsoon year and of the 30–60 day cycle during a drought year.

The latter cycle is known as the Madden-Julian oscillation (Madden and Julian, 1972). Observational analyses and modeling studies revealed that there are dominant periods in the tropics and subtropics. Since the publication of that paper, tropical oscillations have been extensively studied with observations (Yasunari, 1979, 1980, 1981; Sikka and Gadgil, 1990, Yoneyama et al., 2013) as well as by means of models (Goswami and Shukla, 1984, Zhu and Hendon, 2015, Wang et al., 2016). The Madden-Julian oscillation has a significant impact on the Indian (Murakami 1976; Yasunari 1979) and Australian monsoons (Hendon and Liebmann 1990). Kessler and McPhaden (1995) for instance suggested that it could play an important role in the onset and development of El Niño events. The relationship between this oscillation and tropical cyclogenesis has also been posited in some works (Maloney and Hartmann, 2000; Mo 2000; Maloney and Hartmann, 2001). Moreover, He et al. (2011) showed that Hadley circulation variability is closely related to Madden-Julian oscillation convection. Goswami and Shukla (1984) suggested that this oscillation in the Hadley circulation is due to the interaction between the internal dynamics of tropical circulation with moist convection; in fact, with constant latent heat included in their model, the quasi-periodic oscillation vanished. This led them to consider that latent heat released during the moist processes can play a fundamental role in the dynamics of this cycle.

Tropical atmosphere dynamics can be explored via complex general circulation models as well as rather simpler models like the axisymmetric ones. Axisymmetric models focus on the main processes occurring in the tropospheric region, capturing the central process of the Hadley circulation. In such models, for example, eddies are not allowed, and all the processes involved in the higher latitudes are not considered. In this paper, an axisymmetric model (Cessi, 1988) is used to perform an analysis of Hadley cell behavior when periodic forcing is applied. The model spin up takes less than 100 days. When the imposed forcing is not variable, the developing circulation results in a steady state with the Hadley cell representing a fixed point solution within the phase space. In this kind of configuration, other atmospheric processes, acting on longer or shorter spatial and temporal scales, like those mentioned earlier, are essentially excluded.

The Hadley circulation is the meridional overturn that develops in response to a temperature that is in radiative-convective equilibrium, and so it is worth analyzing how the model atmosphere behaves when the vertical stratification of equilibrium temperature becomes neutral. Although the equilibrium temperature towards which the model atmosphere evolves is essentially stable, a less stable temperature stratification may occur, example, due to tropospheric heating caused by condensation of water vapor, resulting in a less stable atmosphere. However, it is reasonable to assume that this condition occurs only periodically.

In Section 2 we describe the model we used and how we set up the stratification by modulating the vertical distribution of the temperature so as to periodically reach neutral static stability. In Section 3, we describe how modulation of the vertical stratification by a periodic function has an impact on the time evolution of the stream-function, which exhibits a periodic response of two main frequencies, suggesting that a periodic response is intrinsic to this model when stratification changes periodically. In this section, we also investigate model sensitivity for some parameters and model configuration (solstitial vs equinoctial). In Section 4 we present our conclusions.

## 2 Model equation and applied forcing

This paper employs the full model described in Cessi (1998), who studied its analytic and numerical solutions via a power series expansion in the Rossby number $R$ of the prognostic variables, the zonal momentum M, the potential temperature $\theta$, and the stream-function $\psi$. The model follows the same hypothesis of the classical axisymmetric models such as those used by Held and Hou (1980) and Fang and Tung (1999).

The horizontal coordinate is defined as $y = a \sin \phi$, from which we have

$$c(y) = \cos \phi = \sqrt{(1 - y^2/a^2)}, \tag{1}$$

where $a$ is the radius of a planet having a rotation rate $\Omega$, with an atmosphere height $H$. The model is similar to the Held and Hou model, but it prescribes a horizontal and vertical viscosity, respectively equal to $\nu_H$ $\nu_V$. The prognostic variables are the angular momentum $M = \Omega a c^2 + u c$, where $u$ represents the zonal velocity; the zonal vorticity $\psi_{zz}$ where the meridional stream-function $\psi$ is defined by

$$\partial_y \psi \equiv w,$$
$$\partial_z \psi \equiv -cv, \tag{2}$$

and the potential temperature $\theta$ that is forced towards a radiative–convective equilibrium temperature $\theta_E$.

Starting from the dimensional equations of the angular momentum, zonal vorticity and potential temperature, we obtain a set of dimensionless equations. The new equations are non-dimensionalized using a scaling that follows Schneider and Lindzen (1977), but the zonal velocity $u$ is scaled with $\Omega a$. A detailed description can be found in Cessi (1998).

The non-dimensional model equations are

$$M_t = \frac{1}{R} \left\{ M_{zz} + \mu \left[ c^4 \left( c^{-2} M \right)_y \right]_y \right\} - J(\psi, M), \tag{3a}$$

$$\psi_{zzt} = \frac{1}{(R^2 E^2)} y c^{-2} \left( M^2 \right)_z - \frac{1}{c^{-2}} J \left( \psi, c^{-2} \psi_{zz} \right)$$
$$+ \frac{1}{(RE^2 c^{-2})} \theta_y + \frac{1}{(Rc^{-2})} \left[ c^{-2} \psi_{zzzz} + \mu \psi_{zzyy} \right], \tag{3b}$$

$$\theta_t = \frac{1}{R} \left\{ \theta_{zz} + \mu \left[ c^2 \theta_y \right]_y + \alpha \left[ \theta_E(y, z) - \theta \right] \right\} - J(\psi, \theta). \tag{3c}$$

where the term $J(A, B) = A_y B_z - A_z B_y$ is the Jacobian.

The thermal Rossby number $R$, the Ekman number $E$, the ratio of the horizontal to the vertical viscosity $\mu$ and the parameter $\alpha$ are defined as

$$R \equiv gH\Delta_H / \left( \Omega^2 a^2 \right); \quad E \equiv \nu_V / \left( \Omega H^2 \right);$$
$$\mu \equiv \left( H^2/a^2 \right) \nu_H/\nu_V; \quad \alpha \equiv H^2 / (\tau \nu_V). \tag{4}$$

The term $\alpha$ is the ratio of the viscous timescale across the depth of the model atmosphere to the relaxation time $\tau$ toward radiative-convective equilibrium.

The boundary conditions for the set of Eq. (3) are

$$M_z = \gamma \left( M - c^2 \right), \; \psi_{zz} = \gamma \psi_z,$$

$$\psi = \theta_z = 0 \text{ at } z = 0,$$

$$M_z = \psi_{zz} = \psi = \theta_z = 0 \text{ at } z = 1, \tag{5}$$

where $\gamma = \frac{CH}{\nu_V}$ is the ratio of the spin-down time due to drag to the viscous timescale; the bottom drag relaxes the angular momentum $M$ to the local planetary value $\Omega a c^2$ through a drag coefficient $C$. The model forcing is represented by an equilibrium potential temperature $\theta_E$ defined by

$$\theta_E = \frac{4}{3} - y^2 + \frac{\Delta_V}{\Delta_H} \left( z - \frac{1}{2} \right). \tag{6}$$

A simulation with the forcing given by Eq. 6 leads to a stationary situation where the Hadley cell is a fixed point for the system of Eq. (3). We refer to this simulation as the control experiment. The parameters used in the simulations we describe in this paper are shown in Table 1.

In order to change vertical stratification we can define $\theta_E$ as an exponential function of z:

$$\theta_E = \frac{4}{3} - |y|^n + \frac{\Delta_V}{\Delta_H} \left( z^k - \frac{1}{2} \right).$$

This approach was used by Tartaglione (2015), who used constant $k$ values, to investigate the role of vertical stratification on the strength and position of the Hadley circulation. The response of the model atmosphere to vertical stratification will be defined by variations of the exponent $k$ in the forcing equation as a function of time. In general, the vertical stratification can change the intensity of the Hadley cell only when the forcing is concentrated on the equator, whereas it loses importance when the forcing is represented by a weak meridional gradient in the equilibrium temperature (Tartaglione, 2015).

Thus, we analyze the response of our model to an equilibrium temperature that moves periodically from a stable stratification to a neutral one; i.e. we allow exponent $k$ of Eq. (6) as a function of time and we assume that $n = 2$. Thus Eq. (6) takes the form:

$$\theta_E = \frac{4}{3} - |y|^n + \frac{\Delta_V}{\Delta_H} \left( z^{1 + sin \frac{2\pi t}{t_0}} - \frac{1}{2} \right). \tag{7}$$

When the exponent is zero the thermal forcing has an adiabatic stratification with no variation of the potential temperature along the vertical direction. The parameter $t_0$ takes a value less than or equal to 90 days, and defines the time in which the forcing becomes adiabatic, forcing the model atmosphere to become neutral. When $t_0$ is small, the main mechanisms involved in the change of the static stability are associated with short time convective events, whereas large values of $t_0$ can be related to longer-lasting meteorological events like monsoons. Whatever process is involved in changing the vertical stratification, we assume in our model that it is periodic.

A cautionary note is necessary. The main role of convection is to bring the atmosphere into a state of neutral vertical stratification, deleting the effects of the unstable layer created by radiative only processes. Thus the dry unstable condition is

**Table 1.** Values of the parameters used in this work

| Parameter | Value | Formula |
|---|---|---|
| H | $8 \cdot 10^3\, m$ | |
| $\Delta_H$ | $1/3$ | |
| $\Delta_V$ | $1/8$ | |
| C | $5 \cdot 10^{-3}\, m^2\, s^{-1}$ | |
| $\nu_H$ | $5\, m^2\, s^{-1}$ | |
| $\nu_V$ | $1.86\, m^2\, s^{-1}$ | |
| $\tau$ | $1.728 \cdot 10^6\, s$ | |
| a | $6.4 \cdot 10^6\, m$ | |
| g | $5 \cdot 10^{-3}\, m\, s^{-2}$ | |
| $\Omega$ | $2\pi/(8.64 \cdot 10^4)\, s^{-1}$ | |
| R | $0.121$ | $gH\Delta_H/(\Omega^2 a^2)$ |
| E | $1.07 \cdot 10^{-3}$ | $\nu_V/(\Omega H^2)$ |
| $\alpha$ | $19.9$ | $H/(\tau \nu_V)$ |
| $\Delta$ | $3/8$ | $\Delta_V/\Delta_H$ |
| $\gamma$ | $25500$ | $CH/\nu_V$ |
| $\mu$ | $4.2 \cdot 10^{-6}$ | $(H^2/a^2) \cdot \nu_H/\nu_V$ |

almost never met in the real atmosphere as there is always an overturn that leads the atmosphere to be statically stable and the most prevalent instability is the conditional stability due to the presence of water vapor. However, the neutral condition of $\theta_0$ represents for this model a less stable condition compared with the $\theta_0$ prescribed in the control simulation (Eq. 6). We parameterize the contribution of the water cycle in the model by means of a time function of $\theta_0$ in such a way that the model atmosphere becomes alternately more or less stable.

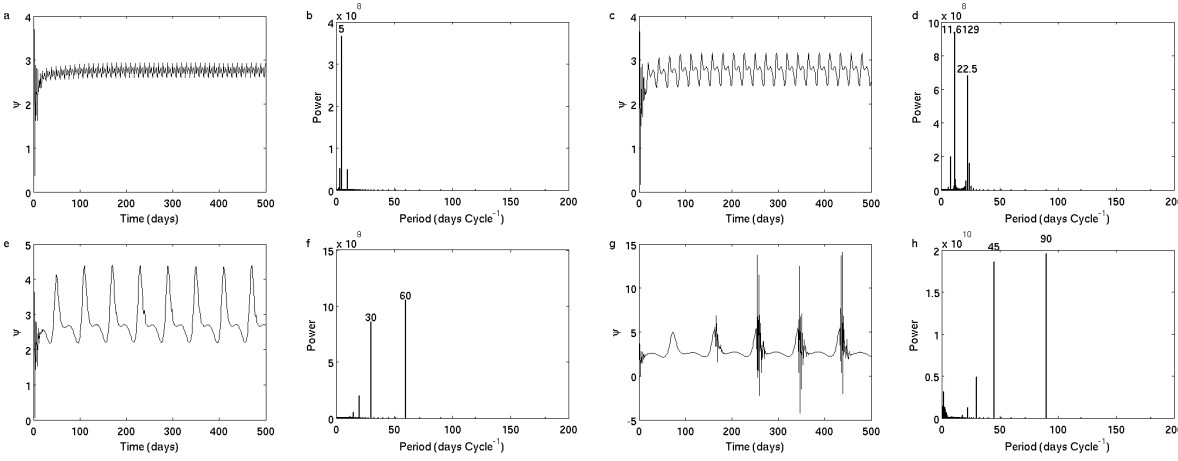

**Figure 1.** Temporal evolution of the stream-function at $3°$ North and 2400 m (a,c,e) and its power spectral density (b,d,f) when $t_0 = 10$ (a,b), 23 (c,d) 60 days (e,f) and 90 days(g,h). The figures on the bars are the cycle periods in days.

## 3   Results

### 3.1   Model response to periodic forcing

Using Eq. (7) as forcing, we obtain a vertical stratification of the equilibrium temperature that becomes periodically neutral. The values of $t_0$ are chosen so as to explore the rate of vertical stratification change, and consequently, the effects on the static

stability. The value of $t_0$ chose lies in the range of 10 to 90 days. The aim of these simulations is to relax the atmosphere to neutral stability, but leave the time average of vertical lapse rate equal to that used in the literature, i.e. equal to one (see Eq. 6).

We now take a closer look at the details in a specific point of the domain, that corresponding to $3°$ N and 2400 m of altitude. The choice of this specific point is based on the fact that the dynamics of this model are essentially equatorial, and that point is in the ascending branch of the cell and its signal in term of spectrum power is higher than that obtained where the value of the

stream-function is a maximum, at about $9°$ N and 1600 m of altitude.

The time evolution of the stream-function, at $3°$ N and 2400 m, and its power spectrum for $t_0 = 10$ days are shown in Figs. 1a,1b. It is easy to see that the evolution resembles a periodic solution with more peaks in the spectrum; however, the dominant frequency is one corresponding to the period of 5 days, i.e., half of the forcing period. As we will see, as the time period increases, the second component related to the time forcing becomes more important. Figures 1c and 1d show the

temporal evolution of the stream-function at a point in the model domain, and the spectral density, for $t_0 = 23$ days. When the forcing period is a prime number, the quasi-periodicity is formally present, as for this value of $t_0$ the two main frequencies are incommensurate. When $t_0 = 30$  days the solution is still periodic with a main period of 30 days, although there is a signal even at 15 days. This behavior is the same at least when $t_0 = 60$ days (Figs. 1e,1f). The peak in the spectrum corresponding to a period of 60 days is higher than that corresponding to the half forcing period.

If we look at the phase space evolution and the plot of couples($\psi$(t);$\psi$(t+$t_0$)), we can see that for both $t_0 = 23$ days and $t_0 = 60$ days (Fig. 2), the solution does not lie on the same curve, but rather assumes a behavior that appears quasi-periodic. As the solution is essentially numeric, this behavior could be related to rounding errors rather than a real quasi-periodic solution.

Time evolution of the solution, for $t_0 = 90$ days, is shown in Figs. 1h and 1g. Looking at the time evolution of the stream-function, the motion is chaotic only when the time is close to multiples of $t_0 = 90$ days, whereas it returns to the steady solution when the stratification $\theta_E$ is far away from the condition of neutral stability. Thus, the slow variation with time, because of the high value of $t_0$, triggers a fast response in the model that allows instabilities to grow faster than those obtained by using small values of $t_0$. Therefore, we observe that there is an unusual sort of slow-fast dynamic. This phenomenon can be viewed as a sort of intermittency with the slow component, which modulates the fast process.

Thus, if the adiabatic forcing is reached with a relatively fast change of the stratification, the solution follows adiabatic forcing, increasing the strength of the circulation (Fig. 3a), which leads to strong subtropical jet streams and stronger easterly winds in the equatorial region (Fig. 3b). As in Fang and Tung (1999), who found stronger circulation when they replaced a fixed sun (equinoctial Hadley cell) with a moving sun, here it is the time-varying stratification stability of $\theta_0$ that causes a stronger circulation with respect to a fixed lapse rate.

Periodicity with two dominant cycles in the model response is interesting in light of the oscillations observed in the tropical atmosphere. Madden and Julian (1971,1972) were the first to show the existence of an oscillation in pressure and winds with a predominant peak in the spectrum at a period of 40-50 days. They also showed that the amplitude of this peak was greater in the tropical station and was weaker in the sub-tropical stations. Yasunari (1979) demonstrated by means of spectral analysis that cloudiness fluctuations have two dominant periodicities: one of about 15 days and another of 40 days. Other studies documented 15–day oscillations within the tropical regions related to monsoons (e.g. Krishnamurti and Bhalme, 1976; Krishnamurti and Ardanuy, 1980; Krishnamurti and Subrahmanyam, 1982). Yasunari (1981) showed that even the 40 day oscillation has some relation to the Asian summer monsoon. Anderson and Rosen (1983) found similar results by using zonally averaged zonal winds. Thus, the features of these oscillations suggest that it may be possible to understand them with a zonally averaged model. Goswami and Shukla (1984) used a symmetric model with hydrology to study the Hadley circulation and found that it has well-defined strong and weak episodes. These oscillations of the Hadley circulation occurred in their model in two dominant ranges of periodicities: one with a period of between 10 and 15 days and another with a period of between 20 and 40 days. Since our model does not include hydrology, this double period has to be related to the internal dynamics of the system. In fact, if periodicity is expected with a time period equal to the forcing time period $t_0$, the response with cycles with more frequencies has to come from the interaction between changing static stability and the internal dynamics of the model. As the changing stratification stability implies a way to simulate the moist convection, our result seems to be in agreement with the findings of Goswami and Shukla (1984). They found that quasi-periodic oscillations of the Hadley circulation were seen only when the moist convective heating was turned on in their model.

The time series of the vertically-averaged stream-function shows that the observed periodic response involves mainly low-level processes. Figure 4a shows the time series of the stream-function averaged over the lower 3200 m, while Fig. 4b shows it averaged over all the domain height. In both cases strong and weak patterns in the stream-function are present, and the strong

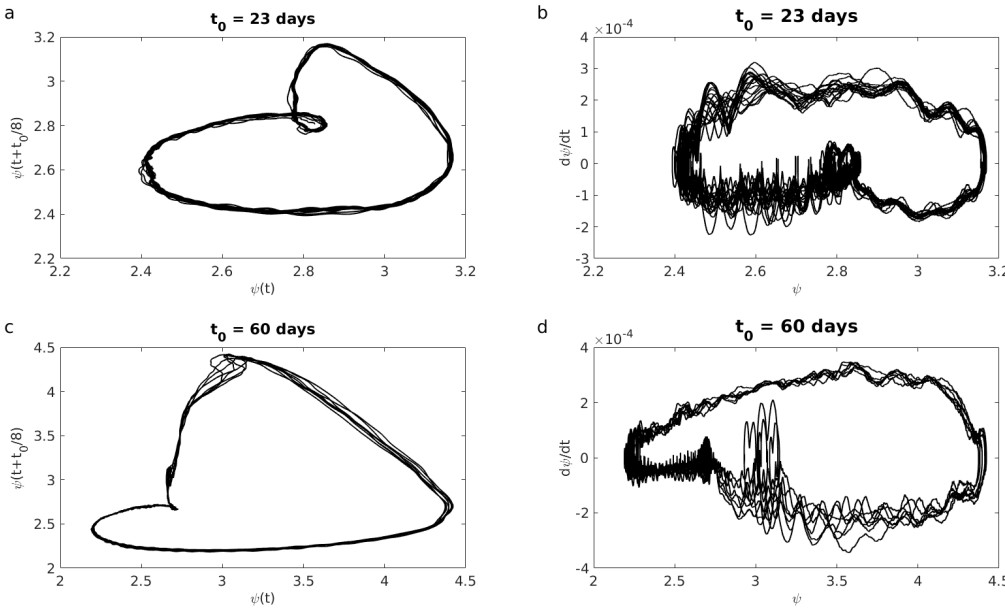

**Figure 2.** Representation of the evolution of the stream-function at $3°$ North and 2400 m $\psi$(t)-$\psi$(t+$t_0$/8) (left panels) for $t_0 = 23$ days (upper panels) and $t_0 = 60$ days experiments (lower panels).

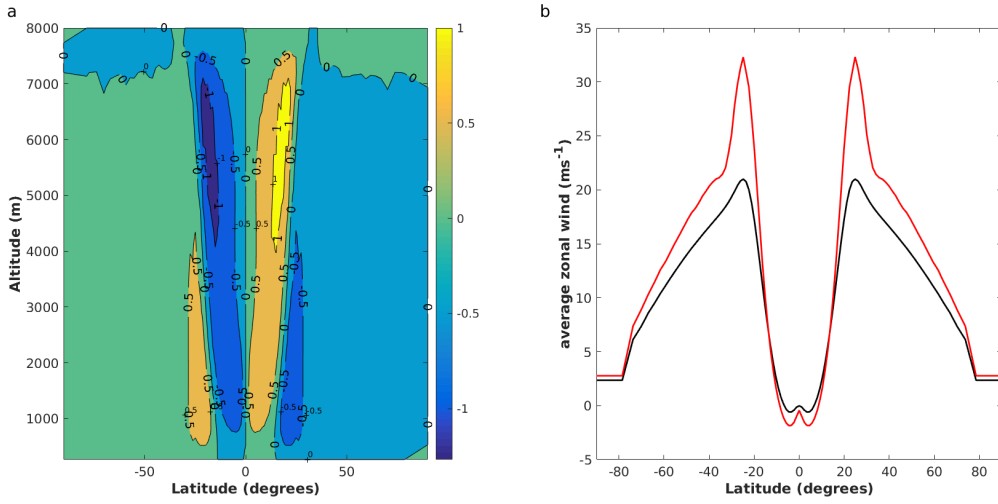

**Figure 3.** Difference between the stream-function (in dimensionless units) of the experiment with $t_0 = 50$ days, and the control experiment (a), the zonal wind U ( in $ms^{-1}$) averaged on the height as a function of latitude y (b), for the experiment $t_0 = 50$ days (in red) and the control experiment (in black) after 500 simulation days.

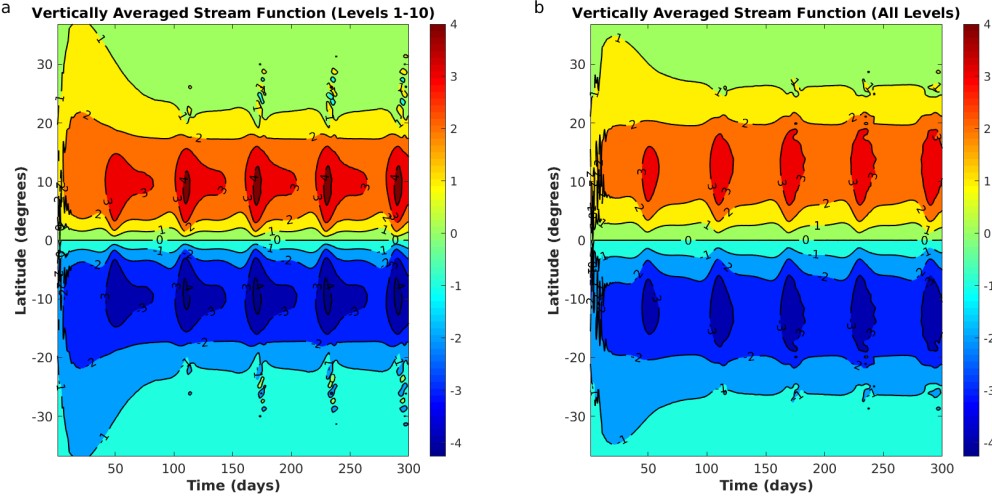

**Figure 4.** Time evolution of the vertically averaged stream-function (non-dimensional unit) over lower levels, up to 3200 m (a) and over the entire height of the domain (b), for the experiment with $t_0 = 60$ days.

episodes of the stream-function at lower levels do not dump suddenly, but persist for a while. However, when averaged over the entire domain time series shows strengthening and weakening of the stream-function that seem to be periodic. This periodic behavior tends to become intermittent whit increasing $t_0$. Higher values of $t_0$ mean that the model atmosphere takes more time to become neutral, while at the same time remaining close to the neutral stability, which induces a perturbation that produces a dramatic increase in the stream-function values.

The interaction of slow parameter variation with the fast rate of motions in the phase space causes a phenomenon known as "dynamic bifurcation" (Guckenheimer and Holmes, 2002). Figure 5 shows a one-dimensional bifurcation diagram, i.e., the differences between two stream-functions at the same point of the domain (3° N and 2400 m of altitude) having a time lag of $t_0$ days, plotted for each value of $t_0$. The outliers, indicating chaotic behavior, start to appear at $t_0 = 64$ days, indicating the presence of spikes in the stream-function values associated with a chaotic solution. Thus, the periodic behavior starts losing force by allowing chaos to emerge when the $z$ exponent approaches zero.

Periodicity is still present, in the sense that chaotic behavior of the model appears periodically. This occurs, for instance, for $t_0 = 90$ days (Figs. 1g,1h). In some respects we can say that the low variability is still governed by periodic oscillation, with the presence of chaotic behavior associated with the neutral static stability of the forcing. Since the transition occurs at $t_0 = 64$ days, which is in the 50-60 day range of the Madden and Julian oscillation, the question arises whether results of this work imply that the Madden-Julian oscillation also has chaotic components. Unfortunately, this is not easy to determine with real data as high frequency components are usually removed in order to study the Madden-Julian oscillations, and because of high non-linearity of the atmosphere where chaotic features are normal.

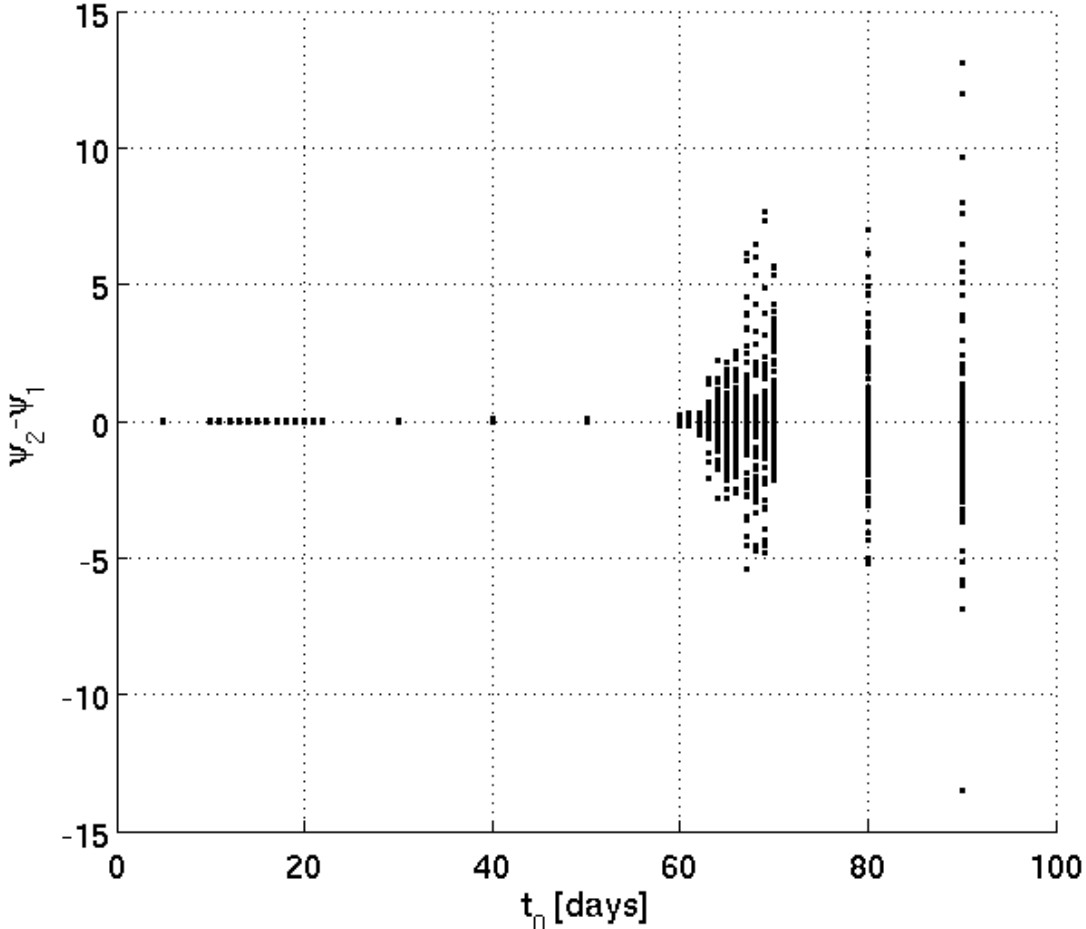

**Figure 5.** The 1-D bifurcation diagram as a function of $t_0$. At a period of 63 days, deviations from periodic behavior start to appear. The plot shows the differences of two stream-functions (in dimensionless units) at the same point of the domain, but with a time lag of $t_0$ days. The value of the vertical viscosity is set to $5\ m^2 s^{-1}$.

It seems that a bifurcation delay might be active when $t_0$ is longer than 63 days. The system fails to notice the onset of instability. This is quite evident in Figs. 1h and 1g, where it is clear that, most of the time, the low frequency (period of 90 days) component is the dominating one; however, a fast component is observed when the chaotic behavior is triggered, a result obtained even with simpler models (e.g. Zaks et al., 2002). Having reached a chaotic state, the system wanders along this state for a time, until it finds itself with the exponent $k$, which defines the slow component of the system, having a value "far enough" from zero that the system goes back to steady solution. At this time, self-modulation is temporarily switched off until the cycle is repeated.

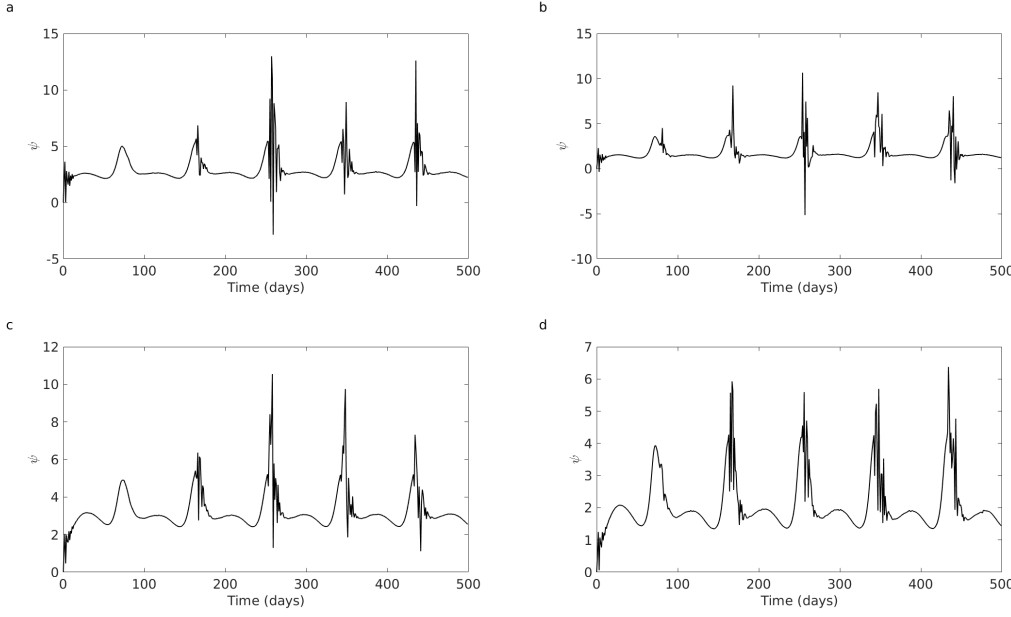

**Figure 6.** As in Fig. 1g, but halving the time step, and increasing the spatial resolution, i.e. halving the horizontal (b), vertical (c) and both (d) grid steps..

We can say that identification of dynamic bifurcations caused by slow variation can, in general, be a problematic task, because of transitions. During these transitions two situations could occur: bifurcations with abrupt change of the attractor size (and in such a case the dynamic bifurcation could be visible), or transitions occurring with changes in the geometry of the chaotic attractor. In the latter case it may be difficult to observe these variations in the short time that the system is in the
5  chaotic state.

## 3.2  Model Sensitivity to time and grid steps

One can argue that the amplification seen in the model solution when $t_0$ equals 90 days may be an artifact of the model. In order to eliminate this possibility, some simulations were performed in order to assess the model sensitivity to time and grid steps. Figure 6 shows the temporal evolution of the stream-function at $3°$ North and 2400 m (as in Fig. 1), obtained by halving
10  the time step (Fig. 6a), doubling the horizontal (Fig. 6b) and the vertical (Fig. 6c) resolutions and halving all the spatial grid steps (Fig. 6d). In all these cases, albeit with some slight differences, there is an explosive increase of the stream-function value at some point, with the most remarkable difference seen when the vertical resolution is increased. There is still periodic strengthening of the Hadley circulation, but its variability is essentially reduced with the negative spikes vanishing. Although not shown here, these results hold when the time step and the spatial grids are even further reduced.

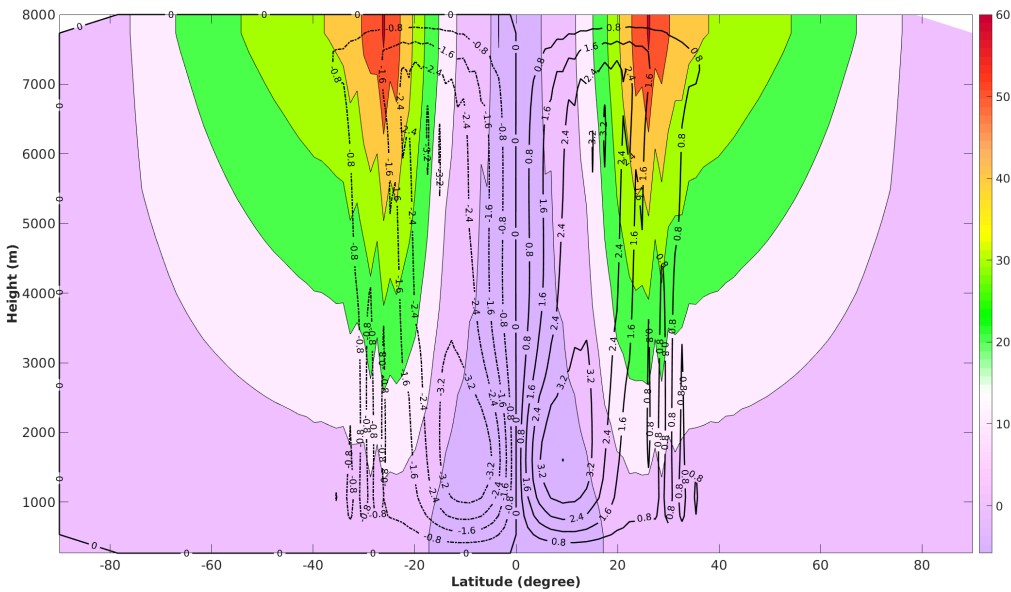

**Figure 7.** Mean stream-function in dimensionless units (contour) and zonal wind in $m^2 s^{-1}$ (color) for the case $t_0 = 90$ days.

The behavior of the numerical solution for large values of $t_0$ has some similarities with the results found by Cessi (1998), who discussed the analytical solution in the case of no stratification. The surface angular momentum departs from the planetary value and is homogenized in the tropical region. The vertically averaged potential temperature is homogenized as well. This homogenization leads to easterly surface winds that are a maximum at the equatorial region, which is dominated by a barotropic core of easterlies that extends upward (see Fig. 7). Westerly winds have higher speed than those obtained with constant stratification. This is already visible in Fig. 2b, where $t_0$ is 50 days. Although there are similarities between the analytic solution found by Cessi (1998) and our numerical solution, they depart from one other, in the stream-function especially, when $t_0 = 90$ days, and even for the case with no stratification. The stream-function behavior starts to appear chaotic, and is actually chaotic when no stratification is imposed (not shown). In fact, the intense chaotic solution with $t_0 = 90$ and no stratification does not blow up, and circulation strength remains limited. With no imposed stratification due to radiative equilibrium, on the one hand the circulation follows potential instability, which tends to lead the isotherms to a vertical position, while on the other hand isotherms are tilted horizontally by the meridional velocity producing a dynamically driven stratification.

Moreover, a second circulation is produced at the poleward edge of the Hadley circulation in the case where $t_0$ equals 90 days (Fig. 7). While this second circulation is unrealistic (we are dealing with a simplified model of tropical atmosphere), it is interesting from a mathematical point of view, as it shows how a numerical solution can differ from the analytic solution, though it was already clear in Cessi (1998) that in absence of stratification second order equations showed a singularity.

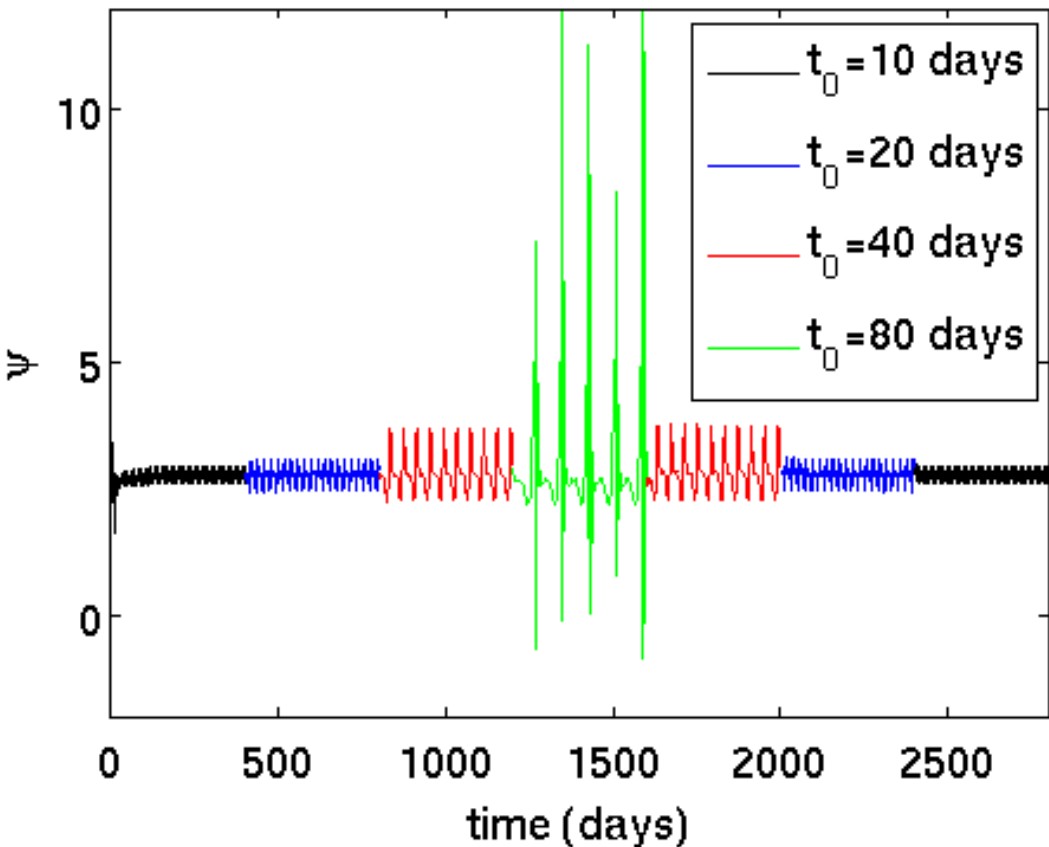

**Figure 8.** Temporal evolution of the stream-function (in dimensionless units) at $3°$ North and 2400 m of altitude for a simulation where $t_0$ changes with time.

To investigate whether history affects the value of the internal state of our model we performed a simulation where we changed the value of $t_0$ during the run from 10 to 80 days, and back again to 10 days (Fig. 8). The model adjusted its response, which is the same of that obtained starting from an initial condition at rest, almost immediately, even though the change of $t_0$ was sudden. This result shows that the model response does not depend on history and on the initial conditions. Moreover the quick adjustment of the model suggests that the fluctuations observed in the simulation are an intrinsic characteristic of it and not an artifact, suggesting that the model, at the least in this form is resilient.

### 3.3 Model Sensitivity to some parameters

Although a parameter uncertainty analysis similar to that presented by Knopf et al. (2006) is outside of the aims of this paper, a sensitivity analysis for some parameters was performed to establish whether found features persist under different conditions.

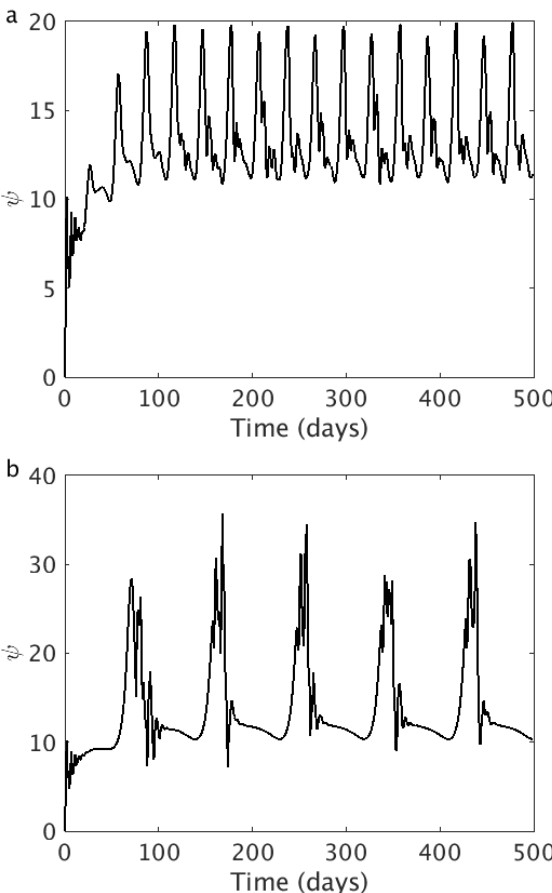

**Figure 9.** The time evolution of stream-function (in dimensionless unit) in the ascending branch of the circulation for t0 = 30 (a) and 90 (b) days with the model in solstitial configuration.

In previous experiments, static stability is forced toward a specific minimum limit, such as neutrality, but whenever it is forced toward values that differ slightly from the neutral condition, the solution behavior remains unaltered.

When the change is significant (e.g., 20% higher or lower than the neutral condition), we need to distinguish the model response as a function of $t_0$. With low values of t0, the periodic solution is independent from the static stability limit. As $t_0$

5   increases, the model becomes more sensitive to the minimum static stability to which the system is forced. In such a case, a more stable minimum prevents the appearance of spikes in the model solution; however, an unstable static stability limit leads to completely chaotic behavior with the appearance of secondary cells far from the equator.

The sensitivity to the parameter $\Delta_H$ is also tested, and the results show that, independently on $t_0$, model response depends on this value. If the parameter is close to 1/3 as set in the previous experiments, we cannot find any difference, but when it

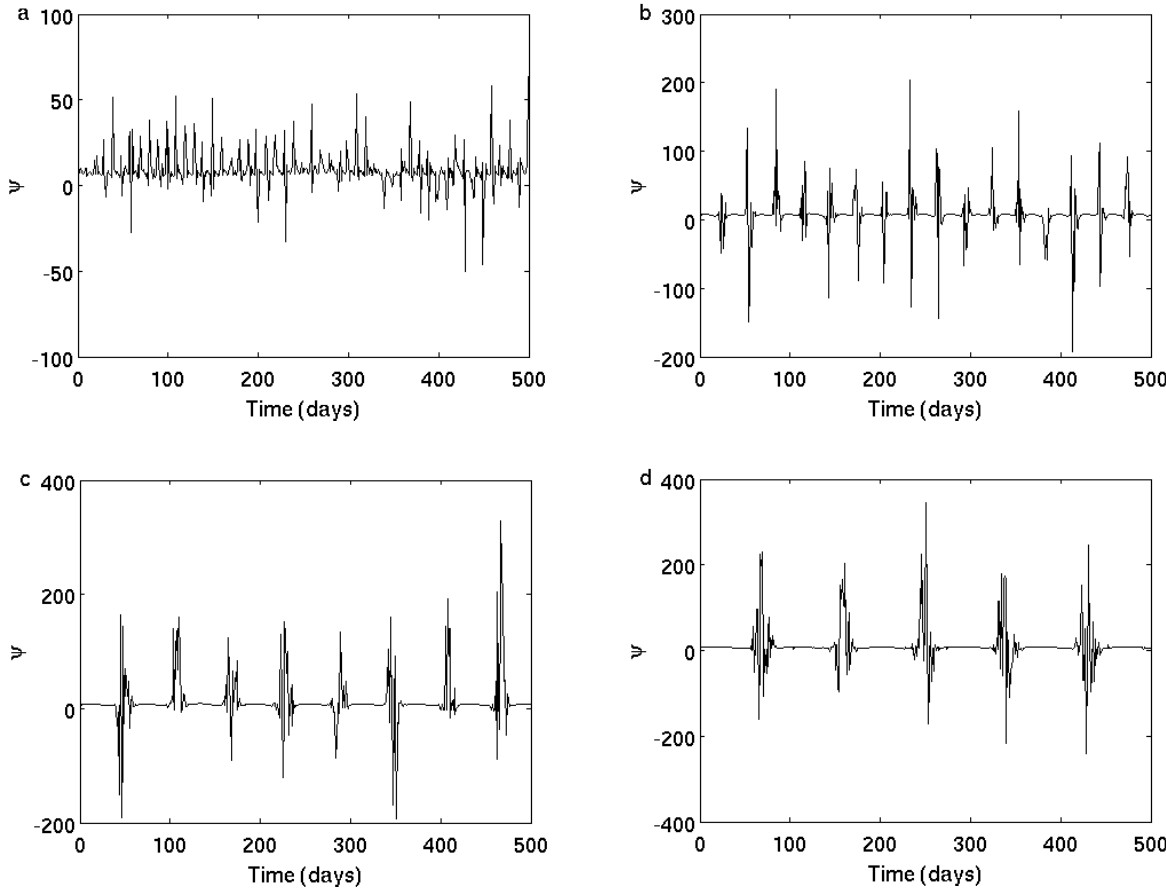

**Figure 10.** The time evolution of stream-function (in dimensionless unit) at $3°$ North and 2400 m for t0 = 10 (a), 30 (b), 60 (c) and 90 (d) when $\nu_V = 0.5$.

departs significantly from that value we notice important changes in the solution. The model response is more stable for low values of $\Delta_H$ and exhibits strongly periodic behaviors. Large values of $\Delta_H$ make the model unstable with a completely chaotic response.

The model configuration has symmetry about the equator. To determine how the model responds to the loss of this symmetry, we performed solstitial experiments with the same parameters used for the equinoctial (symmetric) configuration with maximum heating located in a hemisphere at $6°$ from the equator. Figure 9 shows the stream function value at one point (2400 m of altitude) in the ascending branch of the single cell forms for experiments with $t_0$ values of 30 and 90 days. For low values of $t_0$, the solution is still periodic. When $t_0 = 90$ days, the spikes present in the equinoctial configuration disappear, but the stream function, although noisier, is similar, in term of evolution, to that at $t_0 = 30$ days. The main reason for this change may be the strength of the circulation in the solstitial configuration, which is much greater than that of the equinoctial configuration.

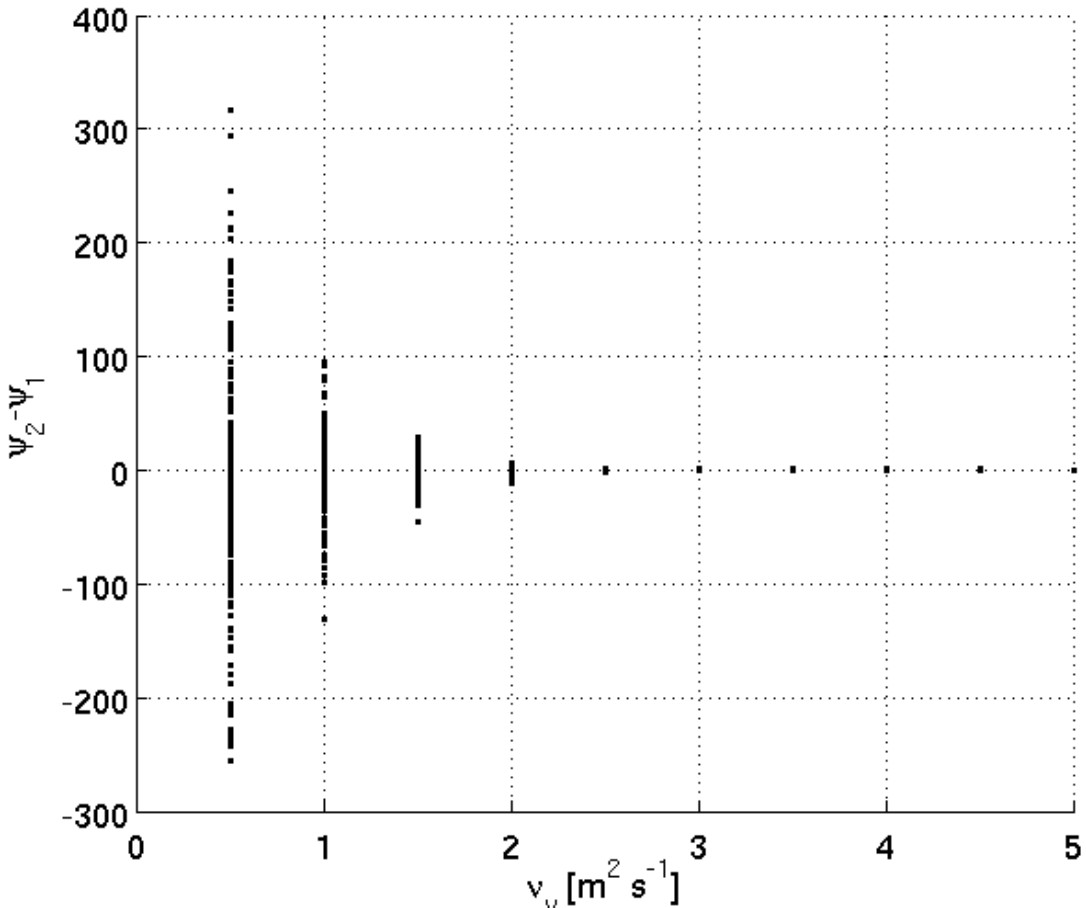

**Figure 11.** The 1-D bifurcation diagram as a function of $\nu_V$ for simulations with $t_0 = 30$ days.

The periodic chaotic behavior is even related to vertical viscosity. It appears even for time periods less than 63 days, when the vertical viscosity is close to zero. When the vertical viscosity is 0.5 $m^2 s^{-1}$, i.e. one tenth of that used previously, the appearance of the chaotic behavior is immediate for small values of $t_0$ , as we can see when $t_0 = 10$ days (Fig. 10a). When $t_0 = 30$ days, the solution is chaotic for almost each of the 30 days (Fig. 10b), and the same occurs when $t_0 = 60$ days (Fig.

5   10c). In such a case, the evolution of the stream-function will have a steady solution with intermittent chaotic behavior when the static stability gets close to the neutral condition, especially for higher $t_0$. Figure 10d illustrates the situation for $t_0 = 90$ days. This representation is only intended to show the effect of the vertical viscosity which must be taken into account when we consider the development and evolution of the stream-function. The high values of the non-dimensional stream-function appear to be unrealistic, but we cannot exclude the possibility transport of momentum and heat from the equator could occur

10   by means of bursts (Majda and Stechmann, 2008). If we wish to estimate for a specific value of $t_0$ at which level of $\nu_V$ this

occurs Fig. 11 gives us insight as to what happens for different values of vertical viscosity, for example when $t_0 = 30$ days. For values less than $2\ m^2 s^{-1}$ the systems begin to exhibit some spikes in the solutions associated with a chaotic behavior of the system. For higher values of $t_0$, the amplification of the model response occurs for higher values of $\nu_V$. On the other hand, high values of vertical viscosity kill the chaotic behavior, making the evolution periodic (not shown).

## 4  Conclusions

We have used a dry axisymmetric model to simulate the Hadley circulation and to investigate the role of a changing stratification of the thermal forcing, which simulates moist convection that alters the static stability of the model atmosphere. The bi-dimensionality of the model prevents the generation of any eastward traveling wave. Hence, in our discussion, the influence of eddies' momentum fluxes is not taken into account. We have shown that the stream-function representing the Hadley circulation in an axisymmetric model can exhibit a periodic behavior when the vertical stratification of the thermal forcing is periodically forced to become neutral. The question that arises is whether this oscillation can be linked with the observed atmospheric fluctuations within the tropical region. Although Madden-Julian oscillation is a three dimensional phenomenon with the development of a Kelvin-Rossby wave (Gill, 1980), it is certainly associated with the evolution of convective anomalies (Hendon and Salby, 1994). It has already been suggested that quasi-periodic oscillation seems to be an intrinsic characteristic of the tropical atmosphere, in accordance with the results of Goswami and Shukla (1984). Bi-dimensional models, while they can give us a framework of basic physics underlying atmospheric processes are quite limited as the real atmosphere is naturally three-dimensional. The findings of this work suggest that if a cyclic process perturbs tropical stratification, the Hadley circulation strength may be periodic with possible bursts when the perturbation time is larger than 60 days. These results must necessarily be compared with observations of the real atmosphere or to the results of three-dimensional models.

If the forcing period is up to 63 days the stream-function evolution shows a periodic behavior. For period forcing longer than 63 days, the slow frequency associated with the forcing period modulate a fast response in the system, generating a chaotic motion that persists for a period of time before returning to a non-chaotic solution. It is not clear whether these high frequency characteristics are actually present in the meridional circulation of our planet. In fact, when we look for periodicities of the order of tens of days in observations, the higher frequency signals are usually removed. Moreover, even though we can detect such signals, it is not easy to associate them with the oscillations caused by a change of static stability, as opposed to other processes. The chaotic dynamics observed in the model could exist in other planets where the vertical stratification takes longer to become neutral. The change of vertical stability that in our model simulates the cycle of large-scale convection might be equivalent to the recharge and discharge of moisture that supports the Madden-Julian oscillation (Zhu and Hendon, 2015). This change of stability can be imagined to control the aggregation process of convection, which allows a bistable equilibrium between moist and dry situations (Raymond and Zeng, 2000, Zhang et al., 2003, Arnold and Randall, 2015). An important aspect is the rate at which this process occurs. As we have shown in the model we consider, when this rate is over 63 days it results in the short term in a chaotic impact on the Hadley circulation strength.

Many parameters control the numerical solution for the model presented in this paper. We have analyzed model sensitivity to a few of these parameters with imposed stratification periodic forcing. The forcing imposes static stability to move toward neutrality, and the model?s response does not change if static stability remains close to neutral, but when it departs from neutrality, there is no change in the model solution for low values of $t_0$. For higher values of $t_0$, however, the model response is strongly affected by static stability. Low static stability prevents chaotic behavior, but higher static stability triggers chaotic motion. Moreover, even $\Delta_H$ is important for higher values of $t_0$ and has the same effects discussed earlier with regard to static stability: low values of $\Delta_H$ minimize chaotic behaviors, whereas high values of $\Delta_H$ tend to trigger very chaotic responses.

The model in the equinoctial configuration shows intermittent chaotic behavior when the forcing period is large because of the importance of the stratification forcing imposed on the model atmosphere. However, this static stability becomes less important in creating large spikes in the stream-function when a solstitial configuration is adopted. With a single strong cell, the circulation is more stable.

The role of friction in the symmetric circulation, driven by a meridional thermal gradient of a fast rotating planet like the Earth, is contradictory. On the one hand it is an essential ingredient to allow a meridional overturn, instead of a strong zonal wind in cyclostrophic balance only. On the other hand, the value must be close enough to zero to allow angular momentum conservation. Although these conditions are met in the model we consider, the presence of a time-varying stratification alters the classic view of a stably stratified vertical temperature gradient. Other than the meridional thermal gradient, this imposed time-varying stratification represents another nonlinear forcing, which amplifies the model response when the vertical viscosity is small, itself representing a source of amplification of the model response in the inviscid case.

*Acknowledgements.* The author wishes to thank two anonymous reviewers who were willing to review this paper. One of them provided useful suggestions to remarkably improve the paper. Uni Research Climate is acknowledged to cover the publication cost.

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
