# Peer review of "Dynamics of the Hadley circulation in an axisymmetric model undergoing stratification periodic forcing"

_Nonlinear Processes in Geophysics, 2016_

## Referee Comment (RC1) · Anonymous Referee #1 · 20 Dec 2016

The forcing from MJO is not stationary because MJO is an eastward moving 'pulse' of cloud and rainfall near the equator that typically lasts for a time period of from 30 to 60 days. I am not convinced that the forcing from MJO can be represented by the axisymmetric model in this paper.

---

## Referee Comment (RC2) · Anonymous Referee #1 · 20 Dec 2016

The forcing from MJO is not stationary because MJO is an eastward moving 'pulse' of cloud and rainfall near the equator that typically lasts for a time period of from 30 to 60 days. I am not convinced that the forcing from MJO can be represented by the axisymmetric model in this paper.

---

## Author Comment (AC1) · 20 Dec 2016

I thank the reviewer for observations made and I agree with the reviewer that real MJO is characterized by an an eastward moving 'pulse'. I said in the paper that this model cannot simulate such a pulse (page 10 line 6), however what the paper underlines is the accordance with results of Goswami and Shukla (1984), where the quasi-periodicity depends on latent heat although the models are different. In any case, even tough the eastward wave exists we cannot neglect the importance of water cycle, that in this model is represented by changing stratification stability.

[Figure]

2016.

---

## Referee Comment (RC3) · Anonymous Referee #2 · 22 Dec 2016

This short paper describes a numerical model study of an axisymmetric atmospheric circulation (somewhat akin to the Earth's tropical Hadley circulation) subject to a periodic forcing that cyclically modifies the static stability. The configuration is not particularly realistic as a representation of the true circulation in the Earth's tropics, but its motivation is justified as an idealized test of an hypothesis that at least some aspects of the observed response of the annular modes of the atmosphere to seasonal and intraseasonal variations of static stability may be determined by the axisymmetric circulation itself, independently of non-axisymmetric waves and eddies.

As such, this is not unreasonable to explore, at least as an academic exercise, and the author sets out a plausible case with reference to a body of recent publications.

Given this motivation, the author's methodological approach, using a reasonably well established axisymmetric numerical model in spherical geometry, seems sound in principle, though does beg a number of questions (some of which are discussed below). A number of numerical experiments are conducted using this model, with a forcing that pushes the static stability periodically towards (precisely!) neutral stability with a period that is varied over a fairly wide range. The results indicate several different regimes, a (so-called) quasi-periodic regime (see my later comments) with periods from around 20-60 days, an intermittently chaotic regime for periods > 60 days and a further regime at forcing periods of <20 days in which the response is strongest at half the forcing period. A simple bifurcation diagram is constructed which illustrates the onset of the intermittent/chaotic regime.

These would seem to be novel and interesting results, but I do have a number of concerns and questions which I enumerate below.

(i) Perhaps a little pedantic, but I would disagree with the classification of the response at moderate forcing periods as "quasi-periodic". The latter term is normally understood to represent a system exhibiting 2 or more incommensurate frequencies - otherwise the attractor is a "simply periodic" limit cycle (or perhaps it really is a torus?). In the cases shown in Fig. 1, the responses show spectral peaks predominantly at the forcing period and integer fractions thereof. These additional peaks in the spectrum look very much as though they are simple harmonics of the forcing frequency omega - i.e. at 2*omega, 3*omega etc. Such components are not incommensurate with the forcing and are therefore not independent. This can be easily tested by constructing phase portraits e.g. using standard delay embedding of the time series shown in Fig. 1 (i.e. plot $\psi(t)$ vs $\psi(t+q)$, where $q$ is a time interval around 1/4 of the forcing period). If the additional frequencies are indeed harmonics, the phase portrait will resemble a simple closed loop, whereas for a genuinely quasi-periodic evolution a topologically more complex object (such as a torus) will result. Examples could perhaps be presented to the reader in an additional figure?

(ii) For periods longer than 60 days, an apparent dynamic bifurcation to an intermittently chaotic behavior appears. The chaotic bursting behavior seems to appear around the phase in the forcing where static stability is becoming very weak? Are you sure the chaotic behavior that is seen is due to a physical instability and not a numerical artifact? I would have like to have seen some kind of evaluation of this (e.g. by varying the spatial resolution and/or an exploration of what kind of motion is occurring when the chaotic bursts appear). Is this due to some kind of symmetric baroclinic instability or some other process? This should be explored more carefully to make sure we are not simply seeing the result of a defect in the model numerics.

(iii) On a related point, the model configuration chosen looks to have a number of singular symmetries, both geometrical (symmetric about the equator) and dynamical (static stability being forced precisely towards neutrality). Is the response and bifurcation sequence dependent on satisfying these symmetries or is the observed behavior generic? This would be important to check, since non-generic responses are unlikely to be observed in a real atmosphere.

(iv) On a similar theme, are these bifurcations to intermittent chaos likely to be observed in a fully 3D atmosphere? This ought at least to be considered and the means to test this discussed in the closing sections of the paper. Are there plans to extend the study to a fully 3D model with similar forcing? This would be the logical next step, but I would be frankly somewhat skeptical that some of these phenomena are robust to interactions with non-axisymmetric flow components.

Overall, therefore, this manuscript raises some interesting issues that might suggest new insights into certain kinds of annular mode variability in the Earth's tropics on intraseasonal timescales. But there are a number of points that need to be clarified before this study can be considered complete. Apart from the reservations outlined above, the work is reasonably clearly and concisely presented, though would benefit from some help from a native English speaker to tidy things up in places. Some examples are given below:

P.2 line 29 "has" instead of "have"

P.3 line 10 Strictly speaking M is the angular momentum per unit mass per unit radius of the planet and has dimensions of velocity.

P.4 line 24 "smaller than"

P.4 line 26 "associated with"

P.4 line 30 "by radiative processes only[?]"

P.5 line 4 "contribution....such a way that..."

P.5 line 5 "alternately" not "alternatively"

---

## Author Comment (AC2) · 24 Jan 2017

My reply to reviewers is attached as supplement

Sincerely

Nazario Tartaglione

Please also note the supplement to this comment:
http://www.nonlin-processes-geophys-discuss.net/npg-2016-59/npg-2016-59-AC2-supplement.pdf

---

## Author Response (AR1)

**Dynamics of the Hadley circulation in an axisymmetric model undergoing stratification periodic forcing**

**NPG-2016-059**

**Reply to reviewers**

I wish to thank both reviewers for their time and observations they did. My answers are in *italics.*

Reply to reviewer 1

The forcing from MJO is not stationary because MJO is an eastward moving 'pulse' of cloud and rainfall near the equator that typically lasts for a time period of from 30 to 60 days. I am not convinced that t the forcing from MJO can be represented by the axisymmetric model in this paper (pag 10 line 4)

*The reviewer is right and I wrote a similar sentence in the first version (page 10, line 4)*

*"We have used an dry axisymmetric model to simulate the Hadley circulation and to investigate the role of a changing stratification of the thermal forcing, as if there were moist convection that alters the static stability of the model atmosphere. The bi-dimensionality of the model prevents the generation of any eastward traveling wave. "*

*Although observations suggest that intra-annual oscillations are better described when it being driven by large-scale disturbance, there are many studies developed conceptual model by considering coupled oscillations in a single column  suggesting the oscillation can be self-contained in a single region.  Adopting simple model allow to understand some aspects that may be particularly relevant to understanding the role of certain features. Further explanations are given in the answer to the reviewer 2. I hope that they clarify the my position..*

Reply to reviewer 2

I wish to thank the reviewer for time and attention he put to read through my paper. He caught the philosophy behind the paper.

1) Perhaps a little pedantic, but I would disagree with the classification of the response at moderate forcing periods as "quasi-periodic". The latter term is normally understood to represent a system exhibiting 2 or more incommensurate frequencies - otherwise the attractor is a "simply periodic" limit cycle (or perhaps it really is a torus?). In the cases shown in Fig. 1, the responses show spectral peaks predominantly at the forcing period and integer fractions thereof. These additional peaks in the spectrum look very much as though they are simple harmonics of the forcing frequency omega - i.e. at 2*omega, 3*omega etc. Such components are not incommensurate with the forcing and are therefore not independent. This can be easily tested by constructing phase portraits e.g. using standard delay embedding of the time series shown in Fig. 1 (i.e. plot (t) vs (t+q)), where q is a time interval around 1/4 of the forcing period). If the additional frequencies are indeed harmonics, the phase portrait will resemble a simple closed
loop, whereas for a genuinely quasi-periodic evolution a topologically more complex object (such as a torus) will result. Examples could perhaps be presented to the readerin an additional figure?

*The reviewer is not pedantic, the reviewer is right, the forcing is not truly quasi-periodic and I was undecided until the end on which term to use. I noticed that when the forcing period is a prime number the response might be classified 'formally' as quasi-periodic, in fact I added the periodogram for the case t0=23 days, how it may be seen the dominant frequencies are 22.5 and 11.625, although close to 23 and 11.5 they are not commensurate. At this point even this case could be classified periodic. I added an additional figure (new Fig. 2) as suggested by the reviewer. Description of new Fig. 2 is on page 7 line 18 of the marked-up manuscript. I was very careful to classify any motion as quasi-periodic.*

2.) For periods longer than 60 days, an apparent dynamic bifurcation to an intermittently chaotic behavior appears. The chaotic bursting behavior seems to appear around the phase in the forcing where static stability is becoming very weak? Are you sure the chaotic behavior that is seen is due to a physical instability and not a numerical artifact?

I would have like to have seen some kind of evaluation of this (e.g. by varying the spatial resolution and/or an exploration of what kind of motion is occurring when the chaotic bursts appear). Is this due to some kind of symmetric baroclinic instability or some other process? This should be explored more carefully to make sure we are not simply seeing the result of a defect in the model numerics.

*As the reviewer can see the formulation of the forcing lead a weaker static stability periodically for any period considered, however, a weaker static stability alone is not the only condition to trigger a burst in the energy transfer to higher latitudes, the other condition is that has to persist for a while.*

*In order to satisfy the reviewer request, a sensitivity study was performed. Four experiments were designed and run:*

*1) Halving time step but with unchanged horizontal and vertical resolution.*

*2) Halving the horizontal resolution*

*3) Halving the vertical resolution*

*4) Halving the horizontal and vertical resolution*
*See new Fig. 6*

*I included a description of what happens in my opinion, even in the light of Cessi (1998) results. (see new Fig. 7)*
*See pages 11, 12 and 13 of the  marked-up manuscript for the discussion about these results.*

3) On a related point, the model configuration chosen looks to have a number of singular symmetries, both geometrical (symmetric about the equator) and dynamical (static stability being forced precisely towards neutrality). Is the response and bifurcation sequence dependent on satisfying these symmetries or is the observed behavior generic? This would be important to check, since non-generic responses are unlikely to be observed in a real atmosphere.

*I am not sure what the reviewer means. What I can say is that even when the model shows a periodic oscillation, this is only in the strength of the stream-function but the pattern remains almost the same. I have studied this model extensively and I can say that it almost impossible that its output diverges from the classical Hadley cell configuration. Hence, the model maintains the symmetry and the features we see are independent of specific details of the model and the forcing.*
*However, the atmosphere, even though restricted to the tropical circulation is more complex than the model adopted here and consequently we cannot transfer all the results to the real atmosphere tout court.*

4) On a similar theme, are these bifurcations to intermittent chaos likely to be observed in a fully 3D atmosphere? This ought at least to be considered and the means to test this discussed in the closing sections of the paper. Are there plans to extend the study to a fully 3D model with similar forcing? This would be the logical next step, but I would be frankly somewhat skeptical that some of these phenomena are robust to interactions with non-axisymmetric flow components.

*I followed the suggestion of the reviewer, adding a paragraph in the conclusion. See page 16 of the marked-up manuscript.*

*Last but not least a professional editor corrected the paper.*

[revised manuscript text omitted]
_{\mathrm{E}} = \frac{4}{3} - |y|^n + \frac{\Delta_{\mathrm{V}}}{\Delta_{\mathrm{H}}}\left( z^k - \frac{1}{2}\right). \tag{7}$$

This approach was used by Tartaglione (2015), who used constant  _k_ values, to investigate the role of vertical stratification on the strength and position of the Hadley circulation. The response of the model atmosphere to vertical stratification will be

**Table 1.** Values of the parameters used in this work

[revised manuscript text omitted]

---

## Author Response (AR2)

REPLY TO REVIEWER

I wish to thank the reviewer for his suggestions and for specifying what he meant in the first review. Reviewer observations are in italics.

*1) I would agree that the phase portraits suggest a singly periodic behavior is dominant, although the small amplitude departures from a single limit-cycle trajectory do appear to show some short timescale periodicity which might be dynamically significant - though this component does not seem to show up in the spectra in Fig. 2. For the 23 day forcing period case the oscillations look to have a period near roughly a 1 day(?) period? But this is probably not a major issue and I am happy to leave this for the author to answer at his discretion.*

1) Actually, I think that in this model, small differences are not dynamically important.

*2) the additional tests carried out are useful in establishing the robustness of the bifurcation that is reported.*

2) First, we must agree on a definition of the concept of robustness. Is it the maintenance of some desired system characteristics despite parameter fluctuations, or is it defined by how the model responds to changes in the architecture of the system (for instance solstitial vs equinoctial configurations)? Because the concept of robustness can be confusing (e.g., Jen 2003), I discussed sensitivity (in order to avoid being misleading) experiments for some parameters in the paper. In particular, I changed the values of $\Delta_H$, which may alter the solution when the changes are large.

*3) I was simply concerned that the model imposed a symmetry in the circulation about the equator which would only really apply either at equinoxes or for a planet whose obliquity was zero. So one could ask whether the same bifurcation would occur if this symmetry were broken e.g. to represent the circulation close to a solstice? Also, how significant is it that the forcing pushes the stratification precisely towards neutral stability at its extreme? What would happen if the minimum stability being forced was either weakly stable or even weakly unstable? These points are again concerned with establishing how robust the behavior might be and therefore how applicable generally the results might be to the real Earth (or other planet).*

3) I performed simulations that depart from the symmetric conditions, and the results are discussed in the paper. Concisely, we can say that the breakdown of symmetry with respect to the equator changes the model response, especially for larger values of t0. Because the circulation is stronger in the solstitial configuration, static stability neutrality is not that important in creating spikes when compared to equinoctial experiments. However, the circulation is still large. Large changes in the static stability limit with respect to neutrality can be influential too, but small perturbations do not change the model's response.

All these new results were added to the section "Sensitivity to some parameters," where the results obtained with different vertical viscosity values were also described.

Jen,  E. Stable or Robust? What's the Difference? Complexity, 8, 12-18.2003. [http://onlinelibrary.wiley.com/doi/10.1002/cplx.10077/pdf]

[revised manuscript text omitted]